# Reviewing Stranger on the Internet: The Role of Identifiability through "Reputation" in Online Decision Making

**Mirko Duradoni** [1,†] , **Stefania Collodi** [1,†] , **Serena Coppolino Perfumi** [2,†] and **Andrea Guazzini** [1,3,*,†]

1   Department of Education, Languages, Intercultures, Literatures and Psychology, University of Florence, 50135 Firenze, Italy; mirko.duradoni@unifi.it (M.D.); collodistefania@gmail.com (S.C.)
2   Department of Sociology, Stockholm University, S-106 91 Stockholm, Sweden; serena.perfumi@sociology.su.se
3   Center for the Study of Complex Dynamics (CSDC), University of Florence, 50121 Firenze, Italy
*   Correspondence: andrea.guazzini@unifi.it
†   These authors contributed equally to this work.

**Abstract:** The stranger on the Internet effect has been studied in relation to self-disclosure. Nonetheless, quantitative evidence about how people mentally represent and perceive strangers online is still missing. Given the dynamic development of web technologies, quantifying how much strangers can be considered suitable for pro-social acts such as self-disclosure appears fundamental for a whole series of phenomena ranging from privacy protection to fake news spreading. Using a modified and online version of the Ultimatum Game (UG), we quantified the mental representation of the stranger on the Internet effect and tested if people modify their behaviors according to the interactors' identifiability (i.e., reputation). A total of 444 adolescents took part in a $2 \times 2$ design experiment where reputation was set active or not for the two traditional UG tasks. We discovered that, when matched with strangers, people donate the same amount of money as if the other has a good reputation. Moreover, reputation significantly affected the donation size, the acceptance rate and the feedback decision making as well.

**Keywords:** stranger on the Internet; reputation effects; online ultimatum game; adolescents

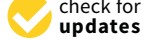



## 1. Highlights

- The propensity to accept an allocation in an online Ultimatum Game is affected by the Donor's reputation.
- Feedback about Donors is affected by their previously acquired reputation.
- A Donor's allocation behavior is influenced by the Receiver's reputation.
- When the reputation is unknown, individuals tend to donate the same amount of money as if the other has a good reputation.

## 2. Introduction

The "stranger on the Internet" effect has been recently presented as the online manifestation of the well-known "stranger on the train" psychological phenomenon [1]. This effect refers to the fact that people disclose significantly more and faster to unknown individuals (i.e., strangers) when further future interactions are not likely [2], which is common on the Internet [3–9]. From a multidisciplinary point of view, the stranger on the Internet effect can also be framed based on game theory. For instance, the generous tit-for-tat strategy in the Prisoners' Dilemma, which encourages being "fair" toward strangers, assuming that they will reciprocate, is the only cooperative Nash equilibrium in specific circumstances [10] and provides higher payoffs in the alternating games [11]. In other words, generous tit-for-tat represents a form of openness towards strangers. While Misoch's work analyzed the phenomenon based on self-disclosure [12,13], the representation of the "stranger" whom individuals interact with has been often neglected by literature.

Given the dynamic development of web technologies, being able to capture and quantify how much strangers can be considered suitable for pro-social acts such as self-disclosure appears fundamental for a whole series of phenomena ranging from privacy protection to fake news spreading [14–17]. For instance, on most social media platforms, users do not actively choose the source of their feed; rather, the platform shows content taken from friends, sources based on past activities, and advertisers who have paid to place their content in the user's feed. The advertisers are usually not known to the target audience (i.e., are strangers) and may target some individuals with malicious intent (e.g., steal personal information, cheat, and share false information).

By employing the Ultimatum Game (UG) [18], it is possible to quantify the representation of the "stranger on the Internet". Indeed, individuals' behavior within the game could be analyzed based on the identifiability of their interactors. The UG is an experimental economics game in which two parties interact, usually in an anonymous way. The first player proposes how to divide a sum of money with the second party. If the second player rejects this division, neither gets anything. If the second accepts, the first gets his/her demand and the second gets the rest. The UG is particularly suitable for our research purposes since it has an empirical robust threshold (approximately 40% of their endowment) on which people rely to allocate resources [19] and so deviation from it may be attributable to our manipulation (i.e., full-anonymity vs. reputation). Moreover, the UG is, for many aspects, "gender-invariant". In the work of Solnick [20], the average offers made did not differ based on gender. Moreover, UG offers seemed to not display a great variability across countries and cultures [21], thus allowing for the greater generalizability of results. A demonstration of a UG session can be seen on Jacob Clifford's YouTube channel for those unfamiliar with the Ultimatum Game: https://youtu.be/_MgMpLTtJA0 (accessed on 27 April 2021).

## 2.1. Reputation and Online Decision Making

Reputation has become a fundamental meter for judging the trustworthiness of sources [22–24]. For instance, reputation affects search result credibility on search engines [25], builds customer trust in e-banking services [26], and influences the consumer decision-making process, particularly in the tourist sector [27]. Indeed, reputation is a cue for understanding which behaviors are accepted within one group. For instance, an e-market "reputational system" deeply influence vendors' behavior [28] by representing, in an economic and perceptively ergonomic way, the system local norm (i.e., to be a trustworthy vendor).

Reputation also affects the decision making of individuals who are not directly identifiable by reputation. Indeed, in e-markets, equally trustworthy individuals realized different exchange volumes according to their reputation [29].

Interestingly, reputation influence upon people's decision making sometimes originates some apparently "irrational" outcomes (e.g., behaviors disconnected from the personal experience). For example, people continue to prefer high-rated partners even if they charge a much higher price for the same good [30]. In other words, people seem willing to accept a worse offer if it comes from a highly reputed member. Moreover, when reputation is not the direct translation of users' behavior (e.g., historical log), reputation attribution can be biased. Indeed, novel empirical findings have highlighted that reputation, once acquired, seems to be maintained over time (i.e., the reputation inertia effect) despite users' actual behavior [31,32].

Given the literature evidence, the paper aimed to quantify prosociality towards strangers, by comparing allocations towards them and those directed to individuals identified by reputation, since reputation is a widely adopted feature to identify people online [33–36]. Moreover, we aimed to quantify how much reputation really counts in attracting offers and inducing acceptance.

### 2.2. Hypotheses

We proposed to our participants an Ultimatum Game, with the reputation visible in some trials and invisible in others to capture behavioral differences towards strangers and individuals identified by reputation. The Ultimatum Game has been widely used to assess people's pro-social behavior in a situation where a second player has some form of power over the first player's behavior [37,38] as in many real-life and online situations where behavioral standards are co-defined (e.g., to be a reliable seller, to offer a service of a given quality, and to adequately protect personal data). The emergence of a standard occurs precisely because an unfair behavior of the first player can be punished with non-cooperation. Introducing a reputation into the game amplifies the ability to make known who respects that standard and who does not. The variation of social behavior based on the interactor reputation allowed us to evaluate how this influences self-disclosure and acceptance dynamics. Based on the main findings presented by the literature, we analyzed how reputation affects people's decision making in an Ultimatum Game, so in the reception and in the donation phases.

Generally, we expect reputation to influence behavior in the reception phase, with a higher reputation broadening the acceptance range and, conversely, a lower reputation narrowing it. In the same way, the level of reputation should affect the feedback; namely, participants will feel more inclined to positively value a subject with an already high reputation, and vice versa. The broader reputation effect, in these cases, should appear when subjects, engaging in an exchange with a counterpart with a high reputation, accept amounts of money below the average threshold (around 40% of the endowment) indicated by literature [19] and evaluate the interaction positively in the feedback phase.

Similarly, we expect reputation to also affect the donation phase; namely, a subject should donate more generously to counterparts with high reputations. This would be in line with indirect reciprocity, which implies that a positive reputation increases the likelihood of being helped by another unrelated person in the future.

In conditions with no reputation, we expect subjects to generally accept on the basis of the average threshold. However, in the donation phase, we could expect social heuristics oriented towards cooperation to take place, so participants will donate more [39].

## 3. Materials and Methods

### 3.1. Sampling

The research was conducted in accordance with the guidelines for the ethical treatment of human participants of the Italian Psychological Association (AIP). The participants were recruited through a completely voluntary census. All the participants (or their legal guardians) signed an informed consent form and could withdraw from the experimental session at any time. The participants were 444 (76 males) with an average age of 15.82 (s.d., 1.30). The ratio between males and females was kept constant in all the experimental conditions. All the participants completed the experiment.

### 3.2. The Conditions and the Game

In order to verify our hypotheses, we developed four different conditions that concerned the presence or the absence of a reputational system (see Table 1).

**Table 1.** Experimental Design. Number of subjects for each condition.

| Experimental Design | | | |
|---|---|---|---|
| | | Reception Phase | |
| | | Reputation ON | Reputation OFF |
| **Donation Phase** | Reputation ON | 111 | 111 |
| | Reputation OFF | 111 | 111 |

Like the original ultimatum game, our game included two phases: donation and reception. Furthermore, the order of the phases was constant; namely, the players played as donors in the first phase and as receivers in the second. Although the participants knew that they were interacting with other players, in reality, the subjects interacted only with our system, which was programmed in order to record the proposals made during the donation phase and to generate offers using an uniform distribution (ranging from 0 to 10 euros) in the reception phase. The system also recorded the players' decisions when they acted as Receivers, and generated a random reputation ranking when necessary for the succession of the different experimental conditions. The probability distribution of reputation was uniform as well. We specify that, in the Reputation ON conditions, the opponent's reputation was displayed from the first turn, since participants were instructed to believe that reputation was achieved by their counterparts during previous sessions acting as Donors, while the player was never characterized by a reputation score.

In the donation phase, participants had to decide how much of their stack (from 0 to 10 euros) they wanted to offer to their counterparts 15 times. The Donors also knew that the Receivers' decision would determine their gain. Indeed, if the Receivers accepted their offers, the resources were split among them according to the Donors' will, while nobody got anything if the Receiver refused the deal. We specified to the subjects that, in our game, the exchanges were asynchronous and delayed in time. In the reception phase, players displayed their interactors as anonymous (as they were themselves), so players could not know if they had interacted with them or not before. In the sessions in which the reputation system was present, Donors could see the reputation (expressed as colored circles and ranging from +5 to −5) acquired by their counterparts. In the reception phase, the players had to evaluate the offers 15 times. Their gain in this phase followed the same rule of the previous one. Like in the donation phase, two reputational conditions were present, so reputation could or could not be visible. After each decision (i.e., accept or refuse), Receivers had to decide about the Donors' behavior by rating them with a plus or a minus. This action was requested to the players to increase reputation credibility. Nonetheless, although the players were instructed to believe that reputation was built up during previous sessions, reputation did not really evolve or change since it was randomly extracted from a uniform distribution, to have approximately the same number of observations for each reputation level.

### 3.3. Procedures

The experiments took place in a computer lab. Before the experimental session started, the experimenters presented the game to the participants. The instructions were read aloud and explained using a PowerPoint presentation. Once the explanation phase ended, the experimenter led the participants to their designated computers. The participants were separated from the others through carton partitions. After completing a brief demographic survey (age, gender, and years of education), the participants obtained the permission to run the game. The experiments lasted a maximum of 30 min.

### 3.4. Data Analysis

First, we verified the preconditions necessary for the inferential analyses of the experiment's data. For the continuous variables that were used, the normality of the distribution was assessed through the analysis of asymmetry and kurtosis values, and their averages and standard deviations were produced. Then, we proceeded to the inferential analyses using a general linear mixed model (GLMM) approach [40] due to the repeated measures structure of the experimental data. GLMM allows seeing, in a robust way, the existence of net combined effects of the variance explained by the factors taken individually.

## 4. Results

In Table 2, the descriptive statistics of the game-related variables are presented.

**Table 2.** Descriptive statistics. In the table are presented means and standard deviation values between brackets for all the game-related variables within each experimental condition.

| Condition | Amount Offered | Acceptance Rate | "Plus" Feedback | Feedback Coherence |
|---|---|---|---|---|
| C1 | 3.40 (1.91) | 0.69 (0.46) | 0.57 (0.49) | 0.76 (0.42) |
| C2 | 3.24 (1.89) | 0.67 (0.47) | 0.59 (0.49) | 0.80 (0.39) |
| C3 | 3.75 (1.89) | 0.65 (0.47) | 0.57 (0.49) | 0.73 (0.44) |
| C4 | 3.42 (1.66) | 0.69 (0.46) | 0.63 (0.48) | 0.80 (0.39) |
| General | 3.48 (1.24) | 0.67 (0.17) | 0.59 (0.19) | 0.78 (0.11) |

*Note:* C1: Rep-ON (Donation) and Rep-ON (Reception); C2: Rep-ON (Donation) and Rep-OFF (Reception); C3: Rep-OFF (Donation) and Rep-ON (Reception); C4: Rep-OFF (Donation) and Rep-OFF (Reception); Amount offered: The quantity of the endowment offered per turn; Acceptance rate: Ratio between the times the Receivers accepted (1) and refused (0) the offer; Plus feedback: The Receivers' rate of positive feedback; Feedback Coherence: The rate of positive feedback towards Donors who offered higher than or equal to 41% of their endowment and of negative feedback to those proposals under this threshold.

### 4.1. Donors' Reputation Affects the Propensity to Accept

We analyzed, through a Generalized Linear Mixed Model, the players' propensity to accept or to reject the Donors' resource allocation. As we could imagine, higher offers were more likely to be accepted by the Receivers. However, also, allocations that came from well-rated Donors were accepted more often with respect to those that had been made by badly rated Donors. No effect of the receiving condition was found in relation to the acceptance behavior (Table 3 and Figure 1).

**Table 3.** Generalized Linear Mixed Models 1. Acceptance dynamics.

| GLMM Best Model for Acceptance Dynamics | | | | |
|---|---|---|---|---|
| | **Model Precision** | **Akaike** | **F** | **Df-1(2)** |
| Best Model | 78.5% | 517.077 | 219.215 *** | 2 (3356) |
| **Fixed effects** | | | | |
| **Factor** | **F** | **Df-1(2)** | **Coefficient ($\beta$)** | **Student *t*** |
| Reputation | 11.174 *** | 1 (3356) | 0.067 | 3.343 *** |
| Amount offered | 436.218 *** | 1 (3356) | 0.546 | 20.886 *** |

*** = $p < 0.001$.

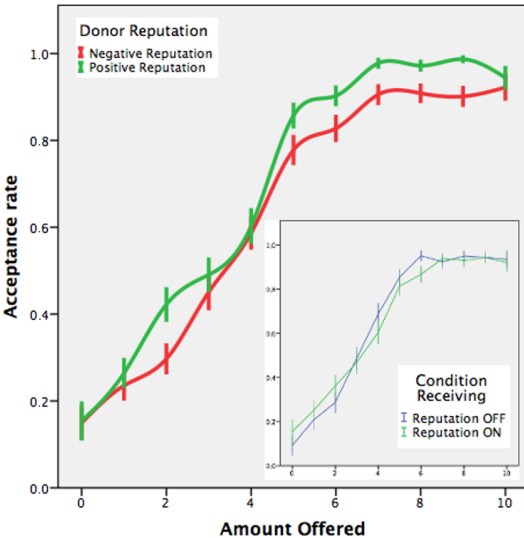

**Figure 1.** Acceptance dynamics with respect to positive and negative reputation of the interactor. In the insert, the acceptance rate trend related to the receiving conditions is represented.

### 4.2. How Do People Evaluate the Donors' Behavior?

Generally, in Ultimatum Games, a "fair" offer is around 40% (i.e., an average of 41.01%) of the amount to share [19]. We use this evidence to define the feedback coherence as follows:

The Receiver provides coherent feedback when he/she gives a plus to Donors' offers higher than or equal to 41% and a minus to those proposals under this threshold. Conversely, the Receiver acts incoherently when he/she gives a plus to Donors' offers below 41% or a minus to those allocations higher than or equal to 41%.

At this point, we investigated which factors could affect the feedback behavior in terms of coherence. In other words, we were interested in assessing whether a change in what is considered "fair" was possible. The results obtained with a GLMM approach are reported in Table 4.

**Table 4.** Generalized Linear Mixed Models 4. Coherence dynamics.

| GLMM Best Model for Coherence Dynamics | | | | |
|---|---|---|---|---|
| | **Model Precision** | **Akaike** | **F** | **Df-1(2)** |
| Best Model | 75.3% | 704.833 | 5.365 *** | 3 (3355) |
| **Fixed effects** | | | | |
| **Factor** | **F** | **Df-1(2)** | **Coefficient (β)** | **Student t** |
| Reputation * Amount Offered | 7.760 *** | 1 (3355) | 0.021 | 2.786 *** |
| Amount offered | 8.857 *** | 1 (3355) | 0.067 | 2.976 *** |

*** = $p < 0.001$.

The Receivers' feedback coherence was influenced by the amount offered by their counterparts. A higher offer was more often considered fair by the Receivers (i.e., coherent feedback), while lower allocations resulted in a fuzzier (i.e., incoherent) feedback behavior. In other words, high offers seem to elicit a greater evaluation similarity (i.e., a positive feedback). Instead, a greater difference in judgments between individuals was observed towards lower offers, with a portion of individuals acting incoherently (i.e., providing positive feedback to offers under the 41% threshold). Furthermore, there is an interaction effect between the amount offered by Donors and their own reputation in relation to the Receivers' feedback coherence. The Receivers were most coherent in that situation with higher offers made by well-rated individuals, while the lowest coherence was found with those offers near the 41% threshold provided by negatively rated interactors (Figure 2).

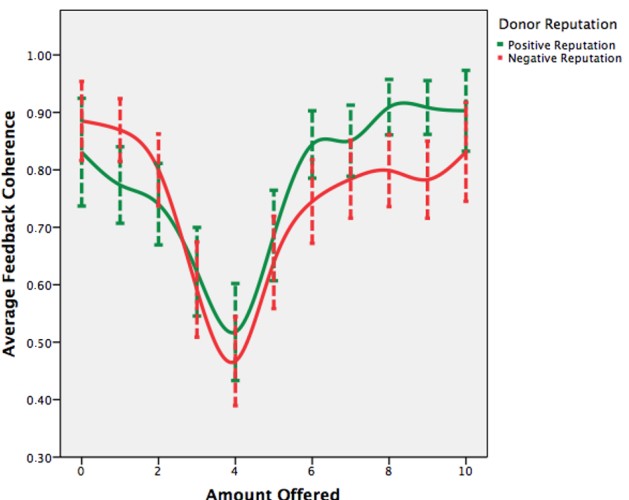

**Figure 2.** Feedback coherence trend in relation to Donors' reputation levels.

Notably, equal offers were treated differently in terms of coherence according to reputation, with those made by well-rated individuals more frequently given coherent feedback.

### 4.3. Players' Feedback Behavior Appears to be Affected by Opponent's Reputation

To investigate the players' tendency to provide positive feedback (i.e., a plus), we proceeded with a GLMM for repeated measures. The results are presented in Table 5.

**Table 5.** Generalized Linear Mixed Models 3. Plus dynamics.

| GLMM Best Model for Plus Dynamics | | | | |
|---|---|---|---|---|
| | **Model Precision** | **Akaike** | **F** | **Df-1(2)** |
| Best Model | 75.5% | 557.493 | 244.047 *** | 2 (3356) |
| **Fixed effects** | | | | |
| **Factor** | **F** | **Df-1(2)** | **Coefficient ($\beta$)** | **Student *t*** |
| Reputation | 15.497 *** | 1 (3356) | 0.068 | 3.937 *** |
| Amount offered | 484.375 *** | 1 (3356) | 0.446 | 22.009 *** |

*** = $p < 0.001$.

More generous allocations were more frequently rewarded with a like from the Receivers. Moreover, the Donor's reputation showed a positive association with the Receiver's probability of feeding back with a like independently from the amount offered. Well-rated opponents more easily acquired further positive feedback, while negatively rated opponents were more frequently evaluated by our participants with a dislike. Therefore, equally "generous" Donors are treated differently in terms of Receivers' positive feedback.

### 4.4. Donation Differences between Conditions

As we can see from Table 6, the average donation in those sessions where the reputation system was enabled was lower. Overall, more "generous" allocations were performed when Receivers were not identified by their reputation. Specifically, when reputation was absent (i.e., totally anonymous interactions), our participants tended to donate the same amount as with an opponent characterized by a good reputation (i.e., a +3 reputational score on a scale ranging between −5 and +5).

**Table 6.** Generalized Linear Mixed Models 3. Effect of the introduction of a reputation system on the donation.

| GLMM Best Model for Donation Dynamics vs. Reputation System | | | |
|---|---|---|---|
| | **Akaike** | **F** | **Df-1(2)** |
| Best Model | 27,604.197 | 57.050 *** | 1 (6685) |
| **Fixed effects** | | | |
| **Factor** | **F** | **Df-1(2)** | **Coefficient ($\beta$)** | **Student *t*** |
| Reputation System (Off) | 57.050 *** | 1 (6685) | 0.352 | 7.553 *** |

*** = $p < 0.001$.

### 4.5. Opponent's Reputation Affects the Amount of Donation

The reputation's influence upon donation decision making was further investigated considering only the donation phases in which the reputation system was provided to our participants.

Figure 3 underlines the existence of a positive relationship between the average participants' donation and the Receivers' reputation. Well-rated Receivers received, on average, a greater portion of the Donor's endowment, while people's donation towards badly rated opponents was smaller (Table 7).

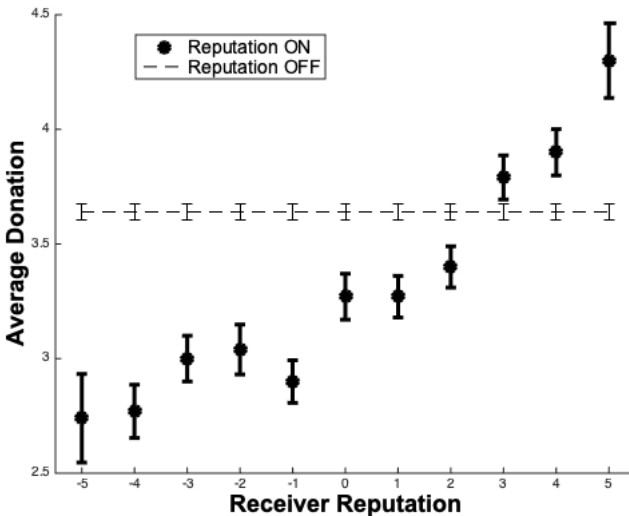

**Figure 3.** Comparison between the average donation trend with respect to Receivers' reputation scores (dots) and the amount offered without the reputation system (dashed line).

Finally, as we can gather from Figure 3, strangers received, on average, the typical donation of positively reputed individuals.

**Table 7.** Generalized Linear Mixed Models 5. Average donation.

| GLMM Best Model for Average Donation Dynamics | | | |
|---|---|---|---|
| | **Akaike** | **F** | **Df-1(2)** |
| Best Model | 13,745.669 | 154.387 *** | 1 (3357) |
| **Fixed effects** | | | |
| **Factor** | **F** | **Df-1(2)** | **Coefficient ($\beta$)** | **Student *t*** |
| Receiver Reputation | 154.387 *** | 1 (3357) | 0.140 | 14.425 *** |

*** = $p < 0.001$.

## 5. Discussion

Participants' donations allowed us to quantify how they depict strangers in terms of reputation. When individuals have to decide how much of their endowment to give to someone else, the reputation of their partner matters. However, when no reputational information was provided by the system, people interacted more generously as Donors. Interestingly, participants donated more on average when their partners had positive reputations (3 out of 5). Without any additional information about their interactors, individuals seem to rely on that automated predisposition towards cooperation individuated by the social heuristics hypothesis' scholars [39] and that is particularly salient from early to mid-adolescence [41]. Since there were no clues about the partners' trustworthiness that could be used to outweigh this heuristic decision-making process, allocations were made more "indiscriminately". Instead, when indirect reciprocity mechanisms are in check, cooperation could be preferentially directed towards those individuals who the group identify as "valuable" members. Giving to an unknown individual (i.e., a stranger) the same amount of resources as to a well-rated person may not be a "mismatch" between humans' evolved psychology and the environment we lived in [42]. Indeed, being generous with unrated individuals could be a successful method for identifying other cooperative individuals and turning them into interaction partners for the future [43].

Overall, our paper contributes to a better understanding of users' behaviors towards online strangers. On the one hand, quantifying users' availability towards strangers may help to prevent the fraudulent exploitation of users [14–17]. Indeed, since we know that

people online tend to trust strangers as if they have a good reputation, new technological solutions and policies are needed to protect users from exploitation. On the other hand, our results could be adopted to ignite patients' self-disclosure in teletherapy settings [44–46]. Self-disclosure is a fundamental aspect of therapy success. The willingness of people to adopt prosocial behavior towards a stranger could be exploited, by appropriately modulating the participants' anonymity, to activate reciprocity dynamics within a therapeutic setting (e.g., an online group therapy) and thus achieve a higher level of self-disclosure.

As for reputation effects, we observed how greatly people rely on reputation for their decision making. Reputation appeared to push individuals to neglect their personal experience (i.e., feedback) and individual preferences (i.e., acceptance) and to adhere to an emerging group standard. This tendency is interpretable within the Social Identity Model of Deindividuation Effects [47,48], which defines the condition needed for conformism to take place online [49].

In our work, we observed that donation, feedback and acceptance behaviors are all preferentially adjusted for well-rated interactors. Moreover, we observed how individuals were susceptible to reputation's influence even when reputation was a "fake" cue. Thus, our work contributes to defining the potential aspects and biases due to reputation dynamics in virtual environments [29], especially in those ones that rely on rating systems such as the one used in e-markets and e-WOM services when people are called to choose a product or service that has been previously evaluated by a community [27,29,50–52].

Independently of size, allocations coming from well-rated interactors were always preferred (i.e., accepted more often). Thus, reputation affected people's acceptance thresholds and pushed them to accept less from individuals with good reputations [30].

Although gender does not usually affect offers in the UG [20], we were not able to exclude the gender effect in our specific sample since our gender ratio within each condition was unfavorable. For this reason, our results may be due to the cooperative and befriending tendency of females regarding everyone, including strangers [53]. For this reason, any future attempt to understand prosocial behavior within the context of an e-game such as UG should really make an effort to recruit almost equal numbers of males and females. Another important aspect to be assessed in future research regards the personal experience that both male and female adolescents gather from UG sessions. The likability of the UG should, in fact, be considered as a possible variable to be accounted for in a future iteration of this experiment. Moreover, since cultural differences in the behavior of responders are possible [21], acceptance-dynamics-related results could be culturally biased. Therefore, future research should assess if our results regarding reputation's influence on acceptance dynamics are robust in other geographical contexts since it has been widely and traditionally used to study people's economic decisions in many parts of the world and in different cultures [54–56]. For what concerns the feedback-related behavior, our results are in line with the previous studies involving widespread feedback systems [31,57]. Independently from the amount offered to the subjects, well-rated individuals had a greater probability of receiving a positive evaluation [31].

Several next steps could be imagined for this research area based on our results. In the end, we only analyzed how the counterpart's reputation affected adolescent players in their choices. What would happen if reputation were placed on them instead of their counterparts is still to be investigated. Moreover, changing the counterpart's rating could be a good path to explore. People in the future will increasingly find themselves interacting with non-human entities thanks to technological advancements. Therefore, assessing how people would behave when matched with social robots [58,59], deep fakes [60], artificial intelligence [61], or other artificial entities rated by reputation would allow estimating how much our results are robust in these scenarios.

## 6. Conclusions

To conclude, people appeared to behave in a prosocial and "open" way when matched with an online stranger (as if the stranger had a positive reputation). Moreover, reputation

appears able to distort the dynamics of a web-based social system. On the one hand, reputation could reduce the effectiveness and robustness of such systems due to potential cascade effects. On the other, reputation "persuasiveness" could be useful for coping with the typical free-riding and social-loafing dynamics of social virtual systems. Therefore, a wise use of identifiability features within virtual environments may help in effectively managing online interaction (e.g., Web Projects) [62] and to contrast some dysfunctional dynamics appearing in public information [63].

**Author Contributions:** Conceptualization, M.D. and A.G.; Data curation, M.D.; Formal analysis, M.D. and A.G.; Investigation, M.D., S.C., S.C.P. and A.G. ; Methodology, A.G.; Project administration, A.G.; Supervision, M.D. and A.G.; Writing—original draft, M.D., S.C. and A.G.; Writing—review & editing, S.C.P. and A.G. All authors have read and agreed to the published version of the manuscript.

**Funding:** This research received no external funding.

**Data Availability Statement:** Not Applicable, the study does not report any data.

**Conflicts of Interest:** The authors declare no conflict of interest.

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
