# Peer review of "Reviewing Stranger on the Internet: The Role of Identifiability through “Reputation” in Online Decision Making"

_futureinternet, doi:10.3390/fi13050110_

Round 1
Reviewer 1 Report
This article suggests a current and attractive topic for the academy. The research is timely and worthwhile. The research problem is clearly defined. The authors provide fresh insight into the field.
I hope you find the following observations helpful:
Structure: Articles should be reformatted according to a standard structure, which is set out in the instructions for authors of the journal (sections are Introduction, Materials and Methods, Results, and Discussions, Conclusion). See the template.
The structure of the paper has to be improved as the discussion wanders off the main topic in several places.
In addition, the paper is a review of a certain topic but the title suggests rather that it is a description of some new method. It needs to be corrected.
Results
The results are explained in a clear and detailed manner.
The art is of different formats, styles, colors, graphics. The paper seems rather to be a copy-paste work than an original study.
Need to revise and check citations in the text and in the references section. I suggest you add references:
- Fedushko S, Peráček T, Syerov Y, Trach O. Development of Methods for the Strategic Management of Web Projects. Sustainability. 2021; 13(2):742. https://doi.org/10.3390/su13020742
- Zakharchenko A, Peráček T, Fedushko S, Syerov Y, Trach O. When Fact-Checking and ‘BBC Standards’ Are Helpless: ‘Fake Newsworthy Event’ Manipulation and the Reaction of the ‘High-Quality Media’ on It. Sustainability. 2021; 13(2):573. https://doi.org/10.3390/su13020573
Overall the work is very good.
So, I think this paper could contribute a lot to the field and could be interesting for the journal.
Author Response
Dear Reviewers,
thank you for the precious and insightful revision. Hereafter we listed the modifications to the paper in order to improve it following your suggestions.
Reviewer comment: Structure: Articles should be reformatted according to a standard structure, which is set out in the instructions for authors of the journal (sections are Introduction, Materials and Methods, Results, and Discussions, Conclusion). See the template.
Authors answer: According to the reviewer’s suggestion, we organized the paper around the template sections. More specifically, we renamed our old “Method” section in “Materials and Methods”, we transformed some parts of the introduction into sub-sections to avoid confusion, and we create the conclusion section.
Reviewer comment: The structure of the paper has to be improved as the discussion wanders off the main topic in several places.
Authors answer: According to the reviewer’s suggestion we revised our discussion section (and the whole paper, for instance in the introduction) to make it more linear and comprehensive. We stressed the aims of the paper and how it contributes to the resolution of the defined problems. These modifications in our opinion made the narration flow more linear, as it should have been from the beginning.
Reviewer comment: In addition, the paper is a review of a certain topic but the title suggests rather that it is a description of some new method. It needs to be corrected.
Authors answer: Thank you for highlighting this issue. As you capture, our goal was to add further quantitative information to an already existing and known effect. For this reason, we modified our title in this way: “Reviewing Stranger on the Internet: The role of identifiability through "reputation" on online decision making”.
Reviewer comment: The art is of different formats, styles, colors, graphics. The paper seems rather to be a copy-paste work than an original study.
Authors answer: First, since it was found redundant, we eliminated the figure regarding the differences in the average donation between the two conditions (old Figure 3). The result was already quite clear with just the related table. Moreover, we removed the title that was on top of one of the figures but not on the others, thus homologating them all on this specific issue. Then we set font-size and font-type equal for all the retained figures. Finally, we removed colors from current figure 3, since they could be misleading. We retained colors just in Figures 1 and 2 since they may help readers interpreting the results about reputation. Moreover, we homologated subsections throughout the manuscript since they appeared different in formatting styles.
Reviewer comment: Need to revise and check citations in the text and in the references section. I suggest you add references:
Fedushko S, Peráček T, Syerov Y, Trach O. Development of Methods for the Strategic Management of Web Projects. Sustainability. 2021; 13(2):742. https://doi.org/10.3390/su13020742
Zakharchenko A, Peráček T, Fedushko S, Syerov Y, Trach O. When Fact-Checking and ‘BBC Standards’ Are Helpless: ‘Fake Newsworthy Event’ Manipulation and the Reaction of the ‘High-Quality Media’ on It. Sustainability. 2021; 13(2):573. https://doi.org/10.3390/su13020573
Authors answer: Thank you so much for providing us with this additional literature that helped us contextualizing our results. We added them in the conclusion section of our paper to highlight the connection between our results and the possible application domains that the reviewer individuated.
Reviewer 2 Report
I have read with interest the paper entitled " Stranger on the Internet: The role of identifiability through "reputation" on online decision making " and I have the following recommendations:
The Abstract needs to be modified. It should focus on explanation of the essence of the problem, specify the aim of the paper, characterise the methodology used and finally sum up the findings.
I see a problem in the structuring of the “Introduction” section. I would recommend definition of the issue in the introduction; state why the given issue is topical and important; sum up the findings on the given topic to date; define the aim of the authors of the paper and state how this paper contributes towards resolution of the defined problems. The theoretical part is not sufficiently supported by bibliographical references.
I recommend modifying the methodology section, for example it is necessary to explain in more details the statistical methods used.
I would also recommend to add some references to “Discussion” section.
Author Response
Dear Reviewers,
thank you for the precious and insightful revision. Hereafter we listed the modifications to the paper in order to improve it following your suggestions.
Reviewer comment: The Abstract needs to be modified. It should focus on explanation of the essence of the problem, specify the aim of the paper, characterise the methodology used and finally sum up the findings.
Authors answer: Thank you for raising this point. We modified the abstract by highlighting what is for us the essence of the problem, i.e., quantify the stranger on the Internet effect to provide insights for privacy protection to fake news spreading. As suggested we also provided more detailed information about the methodology used before summing up the findings in the last part. The following lines were added to the abstract:
“Given the dynamic development of web technologies, quantifying how much strangers can be considered suitable for pro-social acts such as self-disclosure, appears fundamental for a whole series of phenomena ranging from privacy protection to fake news spreading”.
“444 adolescents took part in a 2x2 design experiment where reputation was set active or not for the UG two traditional tasks.”
Reviewer comment: I see a problem in the structuring of the “Introduction” section. I would recommend definition of the issue in the introduction; state why the given issue is topical and important; sum up the findings on the given topic to date; define the aim of the authors of the paper and state how this paper contributes towards resolution of the defined problems. The theoretical part is not sufficiently supported by bibliographical references.
Authors answer: Thank you for allowing us to improve the quality of our manuscript. We enriched our introduction following the reviewer’s suggestions. First, we stressed why the self-disclosure issue is important. Before the hypotheses development, we clarified our aims related to the defined problems. Finally, we add several bibliographical references throughout the introduction. The reviewer will find all the novel parts red-colored.
Reviewer comment: I recommend modifying the methodology section, for example it is necessary to explain in more details the statistical methods used.
Authors answer: We agreed with you about the need for spending few additional lines on the statistical method used. For this reason, we added the following:
“GLMM allows seeing, in a robust way, the existence of combined effects net of the variance explained by the factors taken individually.”
Moreover, we specified that for the continuous variables we produced both mean values and standard deviations.
Reviewer comment: I would also recommend to add some references to “Discussion” section.
Authors answer: To better highlight and discuss our results, we added some additional references. More specifically we inserted in the discussion 1 work concerning prosociality toward strangers in early to mid-adolescence, 3 regarding self-disclosure dynamics in therapy and teletherapy settings, and 4 papers already mentioned in new parts of the introduction about privacy protection and fake news spreading. Finally, in our conclusion, we added two other works regarding the potential impact.
Reviewer 3 Report
April 15, 2021
Review: Stranger on the Internet: The role of identifiability through “reputation” on online decision making.
Comments, questions, and suggestions:
- In page 2 of the paper under the heading “Reputation and online decision making”, the authors cited references regarding this phenomenon as stated in the literature. Everything seems so vague and incoherent without examples. Give examples specifically behavioral examples of the points made in this section.
- In page 2 before the section of “Hypothesis”, it would really be clear and beneficial if the authors DESCRIBE the game. Not everyone is versed or familiar with the game even though it might be popular and very familiar in Italy and Sweden where the authors are from. If this study is to make a contribution in the field, it must be made clear for those who are not familiar to practices made or done in one’s hometown.
- In addition, to be added to item #2 above is to present the rationale why this game serves this study well. Why this game? What aspects of it provides the authors/researchers with the capability to test their hypotheses? Discuss what “reputation” means (another point I make below). How is “reputation” established in the game? Is it already there? Or do players build up the reputation, just like in real life where one’s reputation takes time to develop and mature. Describe the reception phase. Describe the donation phase.
- Sample size of 444 is conveniently distributed among the 4 conditions. However, there were 76 males. That means 368 of the participants are females. Do you not think there is a significant confound in your results and findings when it comes to gender effects? The overwhelming number of females makes it incredibly likely that the results are due to the cooperative and befriending tendency of females about everyone, including strangers. Reputation may or may not affect this befriending nature in that females may perceive reputation as a marker of good character which includes the perception of trust. Males, on the other hand, could potentially see reputation as a marker for a competitor of commodities, resources, mates, etc. Unfortunately, the researchers for this study cannot test this gender difference because the lopsided number of males and females is ridiculous.
- In the procedures section, when participants went to their designated computers, were participants in a room shared with other participants with their computer terminals open and accessible in sight by other subjects? Behavior can be influenced by the presence of others who are seen as coactors or as an audience. To control for this, each participant MUST be working on the game/procedure in isolation without the presence of others.
- At the end of the discussion section, give a couple of examples about APPLICATION of the results to daily events or everyday behavior. What’s the example of reputation in the nongaming reality? Whatever this reputation is conceived to be, does it really work this way out there?
- The other point to make too is that the authors/researchers should make some point about cultural differences in behavior. Maybe in Europe or in Italy or in Sweden, cooperation or good reception is part of the cultural norm. But maybe in South Africa or China or somewhere else, this isn’t the norm. What happens then? Since the results of the study, I suspect, is mainly a reflection of gender differences, I think the results are also a reflection of cultural differences.
- This is the last but just as important a question: What is reputation? If I am perceived by other people to be a kind and generous person, that’s my reputation. If I am perceived by other people as a vain and selfish person, that’s my reputation. Based on the results in this study, does it matter whether it’s a good or bad reputation? Because this study implicates that the answer is that it doesn’t matter even though it really wasn’t tested or examined within the research paradigm.

Author Response
Dear Reviewers,
thank you for the precious and insightful revision. Hereafter we listed the modifications to the paper in order to improve it following your suggestions.
Reviewer comment: In page 2 of the paper under the heading “Reputation and online decision making”, the authors cited references regarding this phenomenon as stated in the literature. Everything seems so vague and incoherent without examples. Give examples specifically behavioral examples of the points made in this section.
Authors answer: Thank you for pointing out this issue. We followed the reviewer request by specifying a little more what the first lines of the sub-section “Reputation and online decision making” mean:
“Reputation has become a fundamental meter to judge the trustworthiness of sources [21–23]. For instance, reputation affects search result credibility on search engines [25], builds customer trust in the e-banking services [26], and influences consumer decision-making process, particularly in the tourist sector [27].”
Reviewer comment: In page 2 before the section of “Hypothesis”, it would really be clear and beneficial if the authors DESCRIBE the game. Not everyone is versed or familiar with the game even though it might be popular and very familiar in Italy and Sweden where the authors are from. If this study is to make a contribution in the field, it must be made clear for those who are not familiar to practices made or done in one’s hometown.
Authors answer: We agreed with the reviewer that describing the game and why it is suitable for our research purposes is essential. We edited the manuscript adding this new part:
“The UG is an experimental economics game in which two parties interact usually in an anonymous way. The first player proposes how to divide a sum of money with the second party. If the second player rejects this division, neither gets anything. If the second accepts, the first gets his/her demand and the second gets the rest. The UG is particularly suitable for our research purposes since it has an empirical robust threshold (approximately40% of their endowment) to which people rely on to allocate resources [19] and so deviation from it may be attributable to our manipulation (i.e., full-anonymity vs reputation). Moreover, the UG is for many aspects “gender-invariant”. In the work of Solnick [20] average offers made did not differ based on gender”.
Reviewer comment: In addition, to be added to item #2 above is to present the rationale why this game serves this study well. Why this game? What aspects of it provides the authors/researchers with the capability to test their hypotheses? Discuss what “reputation” means (another point I make below). How is “reputation” established in the game? Is it already there? Or do players build up the reputation, just like in real life where one’s reputation takes time to develop and mature. Describe the reception phase. Describe the donation phase.
Authors answer:
Thank you for raising these points that allowed us to improve our manuscript. We specified why we chose the UG in the following paragraph in 2.2 section:
“Ultimatum Game has been widely used to assess people's pro-social behavior in a situation where a second player has some form of power over the first player's behavior [37-38] as in many real-life and online situations where behavioral standards are co-defined (e.g., to be a reliable seller, to offer a service of a given quality, to adequately protect personal data). The emergence of a standard occurs precisely because an unfair behavior of the first player can be punished with non-cooperation. Introducing a reputation into the game amplifies the ability to make known who respects that standard and who does not. The variation of social behavior based on the interactor reputation allowed us to evaluate how this influences self-disclosure and acceptance dynamics”.
Moreover, in the introduction, we also added information about the study rationale by answering another reviewer that tangentially responds to some of the points raised.
“Given the dynamic development of web technologies, being able to capture and quantify how much strangers can be considered suitable for pro-social acts such as self-disclosure, appears fundamental for a whole series of phenomena ranging from privacy protection to fake news spreading [14–17]. For instance, on most social media platforms, users do not actively choose the source of their feed; rather,the platform shows content taken from friends, sources based on past activities, and advertisers who have paid to place their content in the user’s feed. Advertisers are usually not known to the target audience (i.e., are strangers) and may target some individuals with malicious intent (e.g., steal personal information, cheat, share false information)”.
Finally, coming up with the questions related to reputation and game mechanics, we specify that a more general description of the UG has been produced for the introduction section as the reviewer requested in a previous point. Instead, the description of the UG used in our study is described in detail in section “3.2. The conditions and the game”. However, since the reviewer raised doubts about how reputation worked in our setting, we modified 3.2 section to clearly state that reputation (i) is only a characteristic of the player’s opponent and so none of the “real” players have one, (ii) it is already there from the first turn and so we had to instruct our participants about the asynchronicity of the game (the instruction was that the reputation that they saw was gained by their opponent while playing as Donor before them), and (iii) reputation do not evolve during the game but it is rather extracted from a uniform distribution to have approximately the same number of observations for each reputation level.
Reviewer comment: Sample size of 444 is conveniently distributed among the 4 conditions. However, there were 76 males. That means 368 of the participants are females. Do you not think there is a significant confound in your results and findings when it comes to gender effects? The overwhelming number of females makes it incredibly likely that the results are due to the cooperative and befriending tendency of females about everyone, including strangers. Reputation may or may not affect this befriending nature in that females may perceive reputation as a marker of good character which includes the perception of trust. Males, on the other hand, could potentially see reputation as a marker for a competitor of commodities, resources, mates, etc. Unfortunately, the researchers for this study cannot test this gender difference because the lopsided number of males and females is ridiculous.
Authors answer: Thank you for highlighting this point. Actually, we do not properly assess this doubt in the text as we should have been but we had in our study. As you noticed the number of males within each condition was quite low and thus we had no sufficient statistical power to capture possible effects of gender on donation. However, since we knew that in the schools we recruited the gender ratio was so imbalanced, we relied on the UG because it is for many aspects “gender-invariant”. In the work of Solnick (2001) average offers made did not differ based on gender. Gender only comes into play affecting donation when connotes the receiver, but that was not our case. The receivers were fully anonymous or identified by reputation. We added a few lines specifying this fact into the just added description of the UG that you requested in the comments above. However, since you wisely raise this doubt and we cannot completely exclude the possibility that in our sample things may be different and gender actually matters for the reasons you exposed, we stressed this as a limitation of our work in the discussion section.
“Although gender does not usually affect offers in the UG [20], we were not able to exclude the gender effect in our specific sample since our gender ratio within each condition was unfavorable. For this reason, our results may be due to the cooperative and befriending tendency of females about everyone, including strangers [45]”.
Ref: Solnick, S. J. (2001). Gender differences in the ultimatum game. Economic Inquiry, 39(2), 189-200.
Reviewer comment: In the procedures section, when participants went to their designated computers, were participants in a room shared with other participants with their computer terminals open and accessible in sight by other subjects? Behavior can be influenced by the presence of others who are seen as coactors or as an audience. To control for this, each participant MUST be working on the game/procedure in isolation without the presence of others.
Authors answer: The reviewer is totally right. We considered this issue in designing the experiment and we used carton partitions to isolate participants from each other.
We specified that in the procedure section as it follows:
“Participants were separated by the others through carton partitions”.
Reviewer comment: At the end of the discussion section, give a couple of examples about APPLICATION of the results to daily events or everyday behavior. What’s the example of reputation in the nongaming reality? Whatever this reputation is conceived to be, does it really work this way out there?
Authors answer: The reviewer raises a very important aspect that we missed in our first draft and that was also noticed by another reviewer. In our discussion, we provided some possible domains in which our results regarding self-disclosure toward possible anonymous interactors (i.e., strangers) can be exploited:
“Overall, our paper contributed to a better understanding of users’ behaviors towards online strangers. On the one hand, having quantifying users’ availability towards strangers may help to prevent fraudulent exploitation of users [14–17]. On the other hand, our results could be adopted to ignite patients self-disclosure in teletherapy settings [43–45]”.
As for examples of reputation dynamics similar to the one “simulated” in our setting, we mentioned in our discussion (and in other already signaled parts of the manuscript) rating systems like the one used in e-markets and e-WOM services (e.g., TripAdvisor) when people are called to choose a product or service that has been evaluated by a community.
“Thus, our work contributes to define the potential aspects and biases due to reputation dynamics in virtual environments [28], especially in those ones that rely on rating systems like the one used in e-markets and e-WOM services when people are called to choose a product or service that has been previously evaluated by a community [26,28,49–51]”.
Reviewer comment: The other point to make too is that the authors/researchers should make some point about cultural differences in behavior. Maybe in Europe or in Italy or in Sweden, cooperation or good reception is part of the cultural norm. But maybe in South Africa or China or somewhere else, this isn’t the norm. What happens then? Since the results of the study, I suspect, is mainly a reflection of gender differences, I think the results are also a reflection of cultural differences.
Authors answer: Thank you once again for giving us the opportunity to strengthen our work. The UG choice served also to account (partially) for this issue. As reported in one meta-analysis there are no big differences in behavior of proposers in UG across geographical regions. So, the results about allocation dynamics appeared quite robust against cultural differences. However, there are possible cultural differences in the behavior of responders. So, the results about acceptance dynamics could be culturally biased. For this reason we added to the paper two parts. The first, is devoted to clarify, once more, why UG is useful in our case to control the possible cultural effect for allocations:
“Moreover, UG offers seemed to not display a great variability across countries and cultures [21], thus allowing for greater generalizability of results”
The second one, is instead devoted to underlining that acceptance results may be culturally specific and thus future research should assess if our results are robust in other countries and cultures:
“Since possible cultural differences in the behavior of responders are possible [21], acceptance dynamics related results could be culturally biased. Therefore, future research should assess if our results regarding reputation influence on acceptance dynamics are robust in other geographical contexts”.
Reviewer comment: This is the last but just as important a question: What is reputation? If I am perceived by other people to be a kind and generous person, that’s my reputation. If I am perceived by other people as a vain and selfish person, that’s my reputation. Based on the results in this study, does it matter whether it’s a good or bad reputation? Because this study implicates that the answer is that it doesn’t matter even though it really wasn’t tested or examined within the research paradigm.
Authors answer: As the reviewer wisely noticed our work did not encompass different types of reputation but only focused on one single type of reputation that refers to fairness and trustability in a very specific scenario (that we hope to have clarified in all the comments above this one). In reality, the reputation seems to have an influence and I hope through our answers the reviewer has been able to appreciate our point. On the one hand, it changes the Donor behavior which in general is very robust to change by about 1.5 euros, passing from a negative reputation to a positive one. The 15% variation is certainly not a negligible effect. On the other hand, although obviously the entity that is offered has a great weight in the receivers’ decision, reputation still plays a role in the acceptance dynamics. If we look at Figure 1, we can see how an offer of 2 euros has almost one and a half times more chance of being accepted if it comes from a counterpart with a good reputation. Clearly this effect is not as great for every possible offer, as we can see from the offer of 4 euros where the difference disappears.
Round 2
Reviewer 3 Report
April 22, 2021
Review of revisions to “Strangers in the Internet…”
The authors did address the comments and questions about the original version of the paper. Good description of the UG game, although it really would be better if there was some demonstration of the game to those unfamiliar with it. I’m not asking the authors to give a demonstration of the game but if they can include a website address or link that shows a demonstration of the game, that will be useful.
Addressing the lopsided number of females and males in the sample size was adequate. The authors should add the note that any attempt to understand prosocial behavior within the context of an e-game such as UG should really make the effort to recruit an almost equal number of males and females. One thing the authors should also include is if the UG game has a reputation in the market as one that has been characteristically favored by male or female adolescents or if it is favored equally by both genders. In the same manner, if the authors could point to some advertised source that can claim that the UG is culturally applicable or not.
In the discussion section, typed up in red ink, on page 9, there are a few sentences in which the authors attempt to apply the results to certain things outside of the study. There is the reference to “fraudulent exploitation of users” and “ignite patients’ self-disclosure in teletheraphy settings”. Okay, where did that come from??? The sudden mention of these application focused conclusions seem so out of place and does not really follow from what was written before it. The authors can devote some time to fix the sentences and communicate the idea about the application value of the results in certain dimensions of behaviors involving interactions between a potential responder and donor, then it might be a better and less painful read.
I’m not sure why there are 3 figures. The figures don’t tell me anything. And the figures don’t say anything different from each other. Where are the means and standard deviation for each of the 4 cells of the 2 x 2 design for each of the dependent variables examined? I’d rather have a table 7 that lists the descriptive statistics of each condition. This table would be more useful and informative than any of the 3 figures.
Overall, the paper is improved. I would try to expand on the implication and application of the results. Doing so would sell this study even further. It would also be a seller if the authors suggest the next set of studies that could potentially follow from this present one.

Author Response
Reviewer comment: The authors did address the comments and questions about the original version of the paper. Good description of the UG game, although it really would be better if there was some demonstration of the game to those unfamiliar with it. I’m not asking the authors to give a demonstration of the game but if they can include a website address or link that shows a demonstration of the game, that will be useful.
Author answer: Accordingly with the reviewer’s suggestion we added an external link regarding a UG demonstration for those scholars that are not into game theory.
“A demonstration of a UG session can be seen from Jacob Clifford’s YouTube channel for those unfamiliar with Ultimatum game: https://youtu.be/_MgMpLTtJA0“
Reviewer comment: Addressing the lopsided number of females and males in the sample size was adequate. The authors should add the note that any attempt to understand prosocial behavior within the context of an e-game such as UG should really make the effort to recruit an almost equal number of males and females. One thing the authors should also include is if the UG game has a reputation in the market as one that has been characteristically favored by male or female adolescents or if it is favored equally by both genders. In the same manner, if the authors could point to some advertised source that can claim that the UG is culturally applicable or not.
Author answer: Thank you for your comment. We actually borrowed from the reviewer his/her words and we edited our discussion adding this sentence in blue:
“For this reason, any future attempt to understand prosocial behavior within the context of an e-game such as UG should really make the effort to recruit an almost equal number of males and females”.
We are not sure to have understood the reviewer properly for the rest of the comment and we apologize for that, but we would like to point out that UG is not a proper “game” that adolescents/adults can play with whenever they want. It is still an experimental setting not so easily accessible. So, we have no idea whether boys or girls would like to play it more or less. We can only speculate about it. We know that females are usually more relationship-oriented than males and the UG is a competitive setting. Thus, it is not unlikely that males would appreciate the UG more. In any case, we added the possibility to rate the experience during the UG for both genders as a future perspective and expansion of our work.
“Another important aspect to be assessed in the future research regards the personal experience that both males and females adolescents gather from UG sessions. The likeability of the UG should in fact be considered as a possible variable to be accounted for in future iteration of this experiment”.
Finally, we pointed to some new references to show that traditionally UG has been used in many cultures:
“...since it has been widely and traditionally used to study people’s economic decisions in many parts of the world and in different cultures [54-56]”.
Reviewer comment: In the discussion section, typed up in red ink, on page 9, there are a few sentences in which the authors attempt to apply the results to certain things outside of the study. There is the reference to “fraudulent exploitation of users” and “ignite patients’ self-disclosure in teletheraphy settings”. Okay, where did that come from??? The sudden mention of these application focused conclusions seem so out of place and does not really follow from what was written before it. The authors can devote some time to fix the sentences and communicate the idea about the application value of the results in certain dimensions of behaviors involving interactions between a potential responder and donor, then it might be a better and less painful read.
Author answer: Thank you for raising this point. We agreed that the sentences the reviewer refers to are presented too abruptly. We tried to add some words, as the reviewer suggested, to make the point more straightforward.
We added the following sentences on page 10:
“Indeed, since we know that people online tend to trust strangers as if they have a good reputation, new technological solutions and policies are needed to protect users from exploitation”.
“Self-disclosure is a fundamental aspect of therapy success. The willingness of people to adopt prosocial behavior towards a stranger could be exploited, by appropriately modulating the participants' anonymity, to activate reciprocity dynamics within a therapeutic setting (e.g., online group therapy) and thus achieve a higher level of self-disclosure”.
Reviewer comment: I’m not sure why there are 3 figures. The figures don’t tell me anything. And the figures don’t say anything different from each other. Where are the means and standard deviation for each of the 4 cells of the 2 x 2 design for each of the dependent variables examined? I’d rather have a table 7 that lists the descriptive statistics of each condition. This table would be more useful and informative than any of the 3 figures.
Author answer: We are really sorry, but we ask the reviewer permission to retain the figures. In our honest opinion, those images are informative and actually help some of the readers to better grasp the results. As for the descriptive statistics, we edited table 2 (which already presented the general descriptive statistics) to show data for each condition.
Reviewer comment: Overall, the paper is improved. I would try to expand on the implication and application of the results. Doing so would sell this study even further. It would also be a seller if the authors suggest the next set of studies that could potentially follow from this present one.
Author answer: Thank you for encouraging us to enhance our paper. As for the application/application of the results, we hope that by answering the previous point now the scenario could be clearer. The reviewer also gave us the opportunity to think about possible next steps of our research. The first one is inspired by a previous comment of the reviewer him/herself (i.e., assessing UG game likeability for both genders). Other possible future studies have been presented as follows:
“Several next steps could be imagined for this research area based on our results. In the end, we only analyzed how counterpart's reputation affected adolescent players in their choices. What would happen if reputation would be placed on them instead of their counterparts is still to investigate. Moreover, changing the counterparts connotation could be a good path to explore.
People in the future will increasingly find themselves interacting with non-human entities thanks to technological advancements. Therefore, assessing how people would behave when matched with social robots (58-59), deep fakes (60), artificial intelligence (61), or other artificial entities connotated by a reputation, would allow estimating how much our results are robust in these scenarios”.